# Exploring perceptions and experiences of gender-based violence among women in a refugee camp setting in Uganda—A qualitative study

**Miriam Lukasiak**[1]*, **Jack Palmieri**[1], **Pia Svensson**[1], **Gilbert Tumwine**[1,2,3], **Anette Agardh**[1]

**1** Department of Clinical Sciences, Social Medicine and Global Health, Lund University, Malmö, Sweden, **2** Department of Obstetrics and Gynecology, St. Francis Hospital Nsambya, Kampala, Uganda, **3** Mother Kevin Post Graduate Medical School, Uganda Martyrs University Nkozi, Kampala, Uganda

* miriam.lukasiak@med.lu.se

**Data Availability Statement:** The study is based on personal data for which legal restrictions for accessing the data apply. The data is deposited at

## Abstract

### Background

Gender-based violence (GBV) is an internationally widespread human rights and public health issue, known to be exacerbated and underreported in humanitarian settings and among conflict-affected populations. A combination of factors including increased vulnerability, lack of protection and marginalization are believed to increase the risk for GBV in settings such as displacement and refugee camps. An increased understanding of GBV in these populations is needed to inform and improve future policy changes and interventions. This qualitative study sought to explore women's perceptions and experiences of GBV in a refugee camp setting in Uganda to increase the understanding of the dynamics and risk contexts of GBV in the context of displacement and refugee camps.

### Methods

This was a qualitative study based on individual semi-structured interviews and content analysis. The interviews were conducted during October 2023 with women living in a refugee camp setting in Western Uganda. The participants (N = 13) included female refugees, residing in the refugee camp, above eighteen years of age and who were survivors of GBV.

### Results

Findings showed no easy escape route from gender-based violence, with a high exposure to GBV throughout the refugee experience. The nature of GBV, the perpetrators and risk contexts however seemed to shift throughout the process from conflict to the refugee camp. Increased marginalization and lack of resources compounded by a shift in gender roles in the refugee camp where women seemed to assume the role of the primary provider increased the risk of violence in pursuit of basic needs. Women described extensive intimate partner violence (IPV) in the camp often connected to new gendered power dynamics and

Lund university, Sweden, and access is regulated by the Research and Ethics Committee (REC), St. Francis Hospital, Nsambya, Uganda, according to the terms under which ethical approval was granted. Data requests may be made to Lund University via registrator@lu.se.

**Funding:** The author(s) received no specific funding for this work.

**Competing interests:** The authors have declared that no competing interests exist.

the control of resources. Faced with the struggles of migration, marginalization, and GBV, women displayed various coping mechanisms including rebuilding networks and support systems.

## Conclusions

Our study showed the complexity of GBV in settings such as refugee camps, where various structural and individual changes involved in migration and life in a refugee camp seemed to create new risk contexts for GBV both inside and outside of the household. Interventions across various dimensions including addressing underlying conditions of marginalization and gendered power dynamics are therefore warranted to address GBV in refugee camps. Further research is essential to better understand this complex issue, as well as the perception and effectiveness of services and interventions in place.

## Background

War, violence, persecution, poverty, and natural disasters force millions of people around the world to flee their homes and seek refuge within or outside of their national borders. According to the World Migration Report from 2022, there were 89.4 million people living in displacement (including refugees, asylum seekers, and internally displaced people) globally at the end of 2020, of which 26.4 million were refugees [1]. Data from the UN Refugee Agency (UNHCR) shows that 75% of the world's refugees and other people in need of international protection are hosted by low- and middle-income countries. Among these, Uganda is one of the five largest recipients hosting around 1.5 million refugees, most of whom originate from South Sudan, the Democratic Republic of Congo and Burundi [2]. In Uganda, 82% of the refugee population are women and children and 95% of the refugees reside in refugee settlements spread across the country [3]. There are 12 refugee hosting districts in Uganda, and the government of Uganda has the primary responsibility for the protection and security of the refugees, which is delivered in collaboration with UNHCR, other UN agencies, and international non-governmental organizations (INGOS) [3].

Gender-based violence (GBV) is an internationally widespread human rights and public health issue, with an estimated 30% of women globally having been exposed to sexual and/or physical violence in their lifetime [4]. The issue of GBV remains underreported in many settings [5] and is known to be exacerbated by humanitarian crises [6]. The Inter-Agency Standing Committee (IASC) defines GBV as: "an umbrella term for any harmful act that is perpetrated against a person's will and that is based on socially ascribed (i.e., gender) differences between males and females. It includes acts that inflict physical, sexual or mental harm or suffering, threats of such acts, coercion, and other deprivations of liberty" [6]. More than just a human rights violation, GBV has been shown to have extensive physical and mental health implications [7]. A wide range of physical injuries have been documented as a consequence of GBV, including various chronic-pain syndromes, gastrointestinal disorders, genital injuries [8–10], traumatic gynecological fistulas [11], and an increased risk of STI incidence [12], including HIV [13, 14]. GBV has also been associated with various psychiatric sequelae such as post-traumatic stress disorder and depression [7, 15, 16].

The issue of GBV in conflict and humanitarian crises has been increasingly studied in recent years [17–19]. Many factors in a humanitarian crisis are thought to increase the risk for

GBV, such as increased vulnerabilities due to the lack of state and community protection and services, displacement, disrupted relationships, lack of essential resources, and discrimination [6, 19, 20]. A common form of GBV in conflict is sexual violence. Sexual violence is a well-known weapon of war used as a means of ethnic cleansing, to dehumanize and destabilize communities [21]. However, evidence also shows that less systematic, haphazard sexual violence is also a common feature of many conflicts [22]. Women and girls in humanitarian settings such as in conflict and displacement have also been shown to be exposed to other various forms of violence such as physical violence, sexual exploitation including prostitution and trafficking, early and forced marriages as well as female genital mutilation [14, 22, 23].

In displacement settings such as in refugee camps, intimate partner violence (IPV) is one of the most frequently reported types of GBV [20, 23, 24]. According to figures from the UNHCR, common forms of GBV experienced among the refugee populations in Uganda are psychological abuse, denial of resources, and physical assault [25]. Apart from the risk factors for GBV mentioned above, men's substance use, loss of financial support, and rapid or forced marriages have also been described as risk factors for IPV in displacement [20, 24]. In a qualitative study on drivers of intimate partner violence against women from three refugee camps in South Sudan, Kenya, and Iraq, it was found that stressors such as poverty and unemployment challenged gender norms in the camps and created unmet gender expectations within the family that could trigger confrontation and IPV [24]. Another study from northern Uganda showed an association between men's level of alcohol abuse, women's prior experience of war-related trauma, and increased levels of IPV in a post-conflict setting [26].

Various local and global strategies for tackling different aspects of GBV in humanitarian settings such as refugee camps continue to be developed [3, 6]. Unfortunately, however, GBV is often underreported and help facilities under-used due to a number of reasons such as shame, stigma, distance to facilities, and costs [5, 27, 28]. In a previous study on experiences of gender-based violence among refugee populations in Uganda, detection of and response to GBV were limited by unequal and gendered power relations and control of resources within the households. Due to feelings of dependency on their husbands, women often refrained from reporting or seeking support [29].

Although there is a growing number of studies on GBV in humanitarian settings and displacement contexts such as refugee camps [17–19], understanding the issue of gender-based violence in refugee and displaced populations is still lacking mainly due to the challenges in conducting such studies given the complex and dynamic nature of these settings [30, 31]. In order to make informed policy changes and interventions more knowledge is needed to better understand the nature and forms of such violence, the risk contexts, and perpetrators of violence in these settings [4]. As wars and natural disasters continue to increase, so will migration and the context of camps for displaced persons and refugees. Thus, a better understanding of these contexts and the issue of GBV in these settings is crucial to be able to serve this population better.

The aim of this study was thus to explore women's perceptions and experiences of GBV in a refugee camp setting in Uganda.

## Methods

### Study design

This was a qualitative study based on individual semi-structured interviews and content analysis [32]. Interviews were conducted with women living in a refugee camp setting in Western Uganda during October 2023. Individual semi-structured interviews were considered suitable

due to the sensitive nature of the subject and because they allowed for a more in-depth examination of individual perspectives and experiences [33].

## Study setting

The study was conducted in Kyaka II Refugee Settlement, located in Kyegegwa District in western Uganda. The settlement is divided into nine zones and twenty-six villages [34], and hosts a refugee population of about 123,873 individuals according to figures from September 2023 [35], most of whom are from Congo DRC and Rwanda [34]. Kyaka II is managed by the Ugandan Office of the Prime Minister's Department of Refugees (OPM) and UNHCR, which in turn work with different partner agencies and NGOs in providing services and protection to the population [34]. The GBV operational presence in Kyaka II Refugee Settlement is mainly provided by Alight, International Rescue Committee (IRC), Lutheran World Federation (LWF), United Nations High Commissioner for Refugees (UNHCR), and United Nations Population Fund (UNFPA) [25]. However, other partner organizations such as Transcultural Psychosocial Organization (TPO) and GoodNeighbors International are also providing both response to and prevention management of GBV in the camp.

## Recruitment

A purposive selection of participants was conducted whereby trained GBV service providers working in partner organizations and NGOs invited female survivors of gender-based violence to participate in the study based on the following eligibility criteria: women, self-identifying as refugees or displaced, residing in the camp, age above 18, with experiences of GBV and currently receiving GBV services from UNHCR's implementing partner organizations. Participants were recruited face-to-face to avoid compromising their confidentiality and safety by written invitations or calls. They were recruited either when visiting facilities such as the 'safe spaces' for women and girls at partner organizations or selected by GBV service providers based on eligibility and exclusion criteria following prior contact with GBV support services. The GBV service providers informed the participants of the study verbally when inviting them to participate in the study, after which the time and date for the interview were set. All of those invited participated in the study. In this study the implementing partner organizations who invited the participants were Alight (4 participants), International Rescue Committee (4 participants), GoodNeighbors International (3 participants), and Transcultural Psychosocial Organization Uganda (2 participants). Exclusion criteria were cognitive impairment or if the GBV-service providers deemed participation in the interview could trigger excessively stressful experiences for the participants.

## Interview procedure

Thirteen semi-structured interviews were conducted with women living in the Kyaka II Refugee Settlement. The interviews were conducted in English by the first author (ML) with the help of interpreters. An interview guide was developed to explore: 1) the characteristics of GBV experienced both in conflict and displacement and the perpetrators in those settings; 2) the perceived risk contexts of GBV in these settings. The interview guide was reviewed in collaboration with representatives of the Office of the Prime Minister Uganda before conducting the interviews. The interviews were recorded and later transcribed verbatim. The interviews were between 30–45 minutes long, and there were no repeat interviews. The data was encrypted and stored safely.

All interviews apart from one were held in the facilities of the partner organizations, so called "safe spaces" for women and girls where counselling, female group meetings, and

vocational training were usually held. The remaining interview was held in the home of the participant as requested by the participant. It was ensured that the participant felt safe in that location. There was no spouse present and according to the participant there were no risks involved with conducting the interview in her home.

Only individual interviews were conducted to ensure the privacy of the participants. The interpreters were all community activists or case workers employed by the respective partner organization. They were all trained GBV service providers, familiar with both the context and issues of the participants. At times the partner organizations required that another staff member, such as a counsellor known to the participant, be present to ensure a safe space for the participant and provide counselling if needed. The first author (ML) was of female gender, medical doctor and researcher, and has previous experience of providing medical assistance to survivors of gender-based violence. All apart from two interpreters were female. Field notes were conducted by a female research assistant employed at the Ministry of Gender Labour and Social Development in Uganda. The two male interpreters were both trained GBV service providers with extensive experience in providing support and services to GBV survivors. It was always verified that the participant was comfortable with the parties present in the room. To ensure confidentiality, all those present were asked to sign a confidentiality form, which was explained to the research participants orally in their native language, prior to the interview. After the interview, participants were informed of the available counseling options and sometimes offered counselling on-site directly depending on their mental state and availability of counselling at that moment. All participants were connected to GBV help services and had previously received or were receiving counselling at the partner organizations.

## Data analysis

Qualitative content analysis as described by Graneheim and Lundman was chosen as the method for data analysis, as it allows for explicit (manifest) as well as implicit (latent) interpretation of data [32, 36]. An inductive approach was used where transcripts were first re-read, then divided into meaning units that were then condensed. Codes were then allocated to the condensed meaning units by the first author. The codes were systematically organized into subcategories and further grouped into categories based on variations, similarities, and differences. The process involved iterative investigation, by moving back and forth between codes, categories and emerging theme. Eventually, an overarching theme was developed, based on the categories and capturing the latent level of the data. The first author had primary responsibility for the analysis with continuous feedback from the co-authors. The transcribed interviews were read twice more by the first author when forming the categories to make sure that the categories aligned with the primary data. Analysis was facilitated by Nvivo 14 (version 14.23.0). The consolidated criteria for reporting qualitative research (COREQ) were followed [37].

## Ethical considerations

Collecting data of this sensitive nature in refugee camps presents ethical challenges particularly due to the lack of stability and security, as well as the heightened vulnerability faced by women and girls in these environments. To address these concerns, we followed the recommendations outlined in the WHO guidance on researching sexual violence in emergency settings [38]. Eligible interview participants were informed of all aspects of the study by the principal investigator; its purpose, process, risks and benefits, voluntary participation, confidentiality, and all safety and security precautions. The participants were informed of their right to withdraw from the study at any time without any explanation or consequences. It was also clarified that

no services received would be affected should the participant choose to participate or not. No payment or incentives were provided for participants. The consent form was written in English and translated verbally to native language by translator. The participants then signed or left a thumb print. Ethical permission for the study was granted by the Research and Ethics Committee (REC), St. Francis Hospital, Nsambya, Uganda. Further approval to conduct the study at Kyaka II Refugee Settlement was given by the Office of the Prime Minister of Uganda.

## Results

Thirteen women, self-identifying as refugees living in Kyaka II Refugee Settlement, participated in the study. Twelve participants came from the Democratic Republic of the Congo (DRC) and one participant came from Burundi. All participants spoke Kiswahili. All participants had experiences of GBV in their country of origin and/or in the refugee settlement (Table 1). All women had or were receiving psychosocial support from the following partner organizations: Alight, IRC, GoodNeighbors International and TPO Uganda.

The findings are presented through one overarching theme: "No easy escape route from gender-based violence", which builds upon four categories. These categories are in turn built upon four to five subcategories, respectively. Together they describe the variations in experiences and perceptions of GBV among women living in the refugee settlement as well as means of building resilience (Fig 1).

### Theme: No easy escape route from gender-based violence

The women describe coming to the refugee camp in hope of peace and stability, fleeing conflict, extensive sexual violence, and unrest in their country of origin. The majority of the participants came from the Democratic Republic of Congo and almost all participants described extensive physical and sexual violence perpetrated by the rebels during the conflict. Coming to the settlement they described being faced with a new set of challenges including marginalization, poverty, and hunger. Although the perpetrators and risk contexts changed, their exposure to gender-based violence remained high. The text below describes the categories and sub-categories in further detail. Participants are referred to using numbers.

**Suffering the health, social, and economic consequences of GBV and forced migration.** Women described the process of migration and the reality of life in the refugee camp filled with new hardships and marginalization. The participants described being exposed to

**Table 1. Characteristics of study participants: Country of origin, time in settlement, experiences of GBV (location).**

| *Characteristics* | *Study participants (n = 13)* |
|---|---|
| **Country of origin** | |
| Democratic Republic of the Congo (DRC) | 12 |
| Burundi | 1 |
| **Time in settlement** | |
| 4–6 years | 11 |
| 10–13 years | 2 |
| **Experience of GBV (location)** | |
| In country of origin | 11 |
| In settlement | 12 |
| Both locations | 10 |

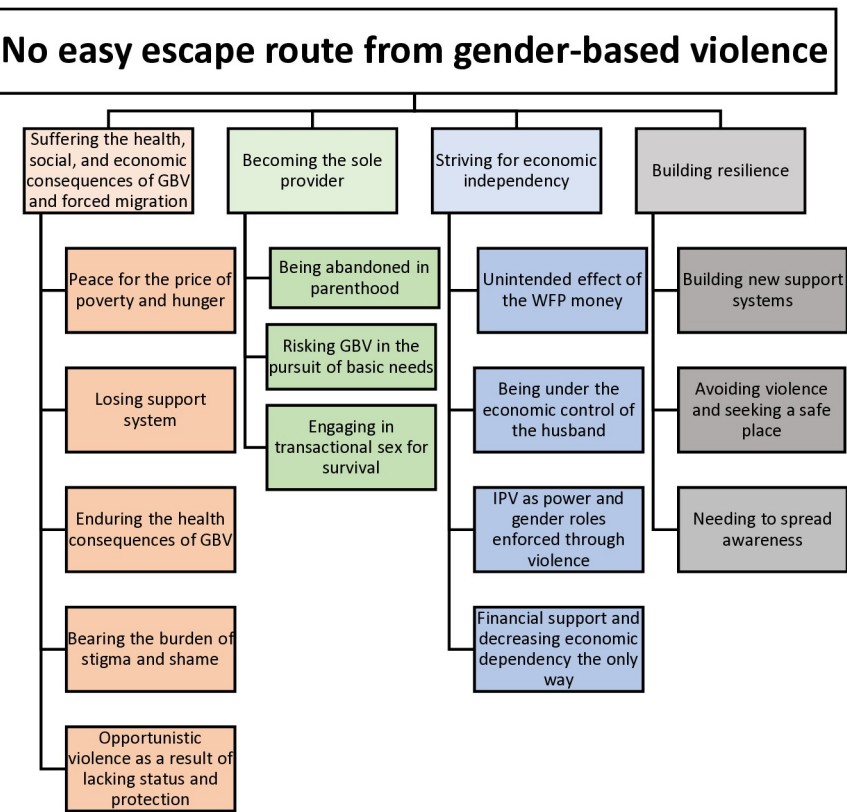

**Fig 1. Overall theme, categories and subcategories describing the experiences and perceptions of gender-based violence among women living in a refugee settlement.**

extensive gender-based violence throughout the process, in conflict and in displacement, and its adverse consequences.

*Peace for the price of poverty and hunger.* The participants described running from war and sexual violence in their country of origin in the hope for peace. Many experienced extensive sexual violence from rebels and strangers in conflict and country of origin. The participants found peace in the settlement in terms of absence of war, but instead faced new hardships such as having to navigate in a new space and facing lack of basic needs. The participants described poverty as being one of the main challenges in the settlement with lack of food, money, and decent living conditions. They described lacking work opportunities and access to education for children due to school fees. The women were worried about the malnourishment of their children and their future if not provided with an education.

*"Life here is good when it comes to peace; we sleep. But when it comes to eating, kids getting an education and having somewhere to sleep it is a problem. Compared to before when we were in Congo, we had somewhere to sleep but no peace. So, the challenge I am facing here is not having anything to eat."*

(IP 8)

The monthly rations provided by the United Nations World Food Program (WFP) were described as scarce and not sufficient to support the family for a month, especially after recent

reductions due to COVID-19. Some even stopped receiving the food support altogether due to recent reductions which made the situation even more dire.

*"Last year WFP was supporting us, but this year they have not supported us and the money they are giving out, the twelve thousand does not cover school fees and food. So, the children are just there now, sitting at home, not going to school."*

(IP 12)

*Losing support system.* Participants described losing their support system and network during conflict and migration. The women expressed feeling alone, not having any family, friends, or relatives in the camp. Many relations were left behind in their country of origin and participants described losing family members, relatives, parents, spouses, and children in brutal killings or abductions during the conflict in Congo. Some expressed feeling alone, not having anyone to get emotional support from and feeling less safe in the camp because of the lack of parents and family to turn to in case of hardships and abuse by husband.

*"...In Congo at least I was near my parents. After that man was mistreating me, pushing and pulling me, I went back home to my parents. My parents took me to the hospital for testing. But then after my parents passed away in war, I travelled to Uganda where it is worse because I don't have any family here. I am alone."*

(IP 10)

*Enduring the health consequences of GBV.* Women expressed various forms in which their health and well-being were affected by past experiences of gender-based violence. Emotional harm, episodes of mental breakdown, recurring thoughts and memories of abuse were some of the ways that women described the effect of GBV on their mental health. Some participants described how their daughters have been traumatized after sexual violence.

*"When my daughter is asleep, she always wakes up shouting "people have come, they want to rape me". Those are the words she always says especially at nighttime."*

(IP 12)

Some participants described becoming infected with HIV and other STIs such as syphilis after sexual violence. Other perceived consequences mentioned were unwanted pregnancies after rape, complications during childbirth, and cases of children dying shortly after delivery as well as difficulties conceiving.

*"I was raped, I got pregnant and infected with HIV. Unfortunately, even the kid I gave birth to...she was also infected and died."*

(IP 8)

A case of rape resulted in severe uterine damage that had to be operated on. Chronic pain, problems with blood pressure, and other physical symptoms were also described.

*"I am not settled... I am having high blood pressure because of these issues. I am having a lot of back pain, and I explained to the doctor that there is something painful inside my eye. I cannot sleep."*

(IP 12)

The participants described being tired and facing mental hardships carrying the weight of all the challenges, poverty, and violence. The women expressed being afraid for their children, feeling worried, having thoughts of suicide, and even resorting to violence towards their children when faced with challenges or failing to provide for the basic needs of the family.

*"When I sit down and think of all the challenges, I think of committing suicide. Especially when the children start asking me for food that I don't have, I think of committing suicide."*

(IP 13).

*Bearing the burden of stigma and shame*. Participants described having to face the social consequences of gender-based violence and its sequelae. Women described social stigma, rumors being spread, and being abandoned, or being unable to find a spouse due to the stigmatizing consequences of gender-based violence such as reproductive difficulties and STIs.

*"So the eldest daughter got a fiancé who wanted to marry her. But when some people here realized that they were going to get married, they told the boy not to marry her because her uterus was removed and so she would never give birth. It is what they told the guy, and the guy decided to leave her. She kept hiding inside because she feared people who knew us from Congo would tell our stories. They knew our stories."*

(IP12)

Speaking of the experiences of gender-based violence was sometimes described as shameful for women and some feared to be ridiculed when sharing.

*"Like if I experience any violence like . . . rape, it is shameful as a woman to come out and talk about it even though it is violence."*

(IP 1)

*Opportunistic violence as a result of lacking status and protection*. In this new state of marginalization, material hardship, and lack of protection, the participants expressed being exposed to haphazard violence by nationals or other members of the community. Gender and power dynamics between migrants and host community, as well as between those with different social status within the camp, left the women feeling exposed and defenseless to opportunistic, random acts of violence.

*"Of course here, people´s status is not the same. There are those who have money. They are the ones who sometimes commit physical violence. Someone can find you there seated, disturbs you. . .then when you try to fight for yourself, the person beats you. And if you talk. . .you don't report because they have money and you don't. They have money to win the case. "*

*(IP8)*

**Becoming the sole provider.**   The participants described assuming the role of the primary provider for their household in the refugee camp. Due to various reasons the women felt left alone in facing the new hardships and lack of basic needs, having to find ways for themselves and their families to survive. In the quest of providing for their families many were exposed to new risk contexts and gender-based violence.

*Being abandoned in parenthood.* Some of the interviewed women had lost their husbands during the conflict in Congo and became single parents. Others felt their husbands 'resigned' as parents, stopped caring and planning for their children, leaving the participants alone in parenthood.

*"Now I am the mother, the father. . .I am everything to these children."*

(IP12)

*"The plans which we had before the man got a little. . .we used to plan for the future together with my husband but not any longer. Especially now. . . supporting my children alone is very difficult for me. Even getting food is difficult for me. I am scared my children will become thieves or get murdered when they are young.*"

(IP11)

Husbands were also described as abandoning or acting out when feeling unable to live up to their expected gender roles, such as providing or protecting their families.

*"And our husbands when they see that they cannot put food into the house they decide to run away and look for prostitution. . .and they leave us at home, they abandon us with the children."*

(IP5)

Others described changing circumstances in which men have no jobs and the women go out looking for casual labor.

*"What I have experienced here in Kyaka, in the camp is. . . When they come here from Congo men have no jobs. And it is the women that go to look for casual labor or to work for food."*

(IP2)

*Risking GBV in the pursuit of basic needs.* The women tried to provide their families with basic needs in various ways. Looking for casual labor or necessities, asking neighbors for help, and savings groups were just some of the methods mentioned. Some managed to create small-scale businesses such as making things they could sell. Loans were also mentioned as a means to get by. Loans often resulted in debt spirals where monthly rations had to be used to pay back loans.

In pursing these basic needs, women were often subjected to gender-based violence. In seeking casual labor opportunities or food, women commonly went to the host community, where many described being subjected to sexual violence. After doing the work required, it was not uncommon for the men from the host community to demand or force sex for the women to receive the payment.

*"Most of us go to the host community when we want to get food and when we reach there, even after providing our casual labor. . .they ask for sex before they give us the reward."*

(IP 6)

Similar cases were described when going to work for the soldiers in the barracks nearby where women sometimes went to work on the plantations. After finishing their work, the women would be raped before being given payment.

*"We go to the plantations of those soldiers in the Kabamba forest, we dig for them, we peel for them, do things. . .after they will get us, they will give us small money. When you are working with them, they just want to rape you. Now if you can run, you run. . .if God is not on your side they will rape you. That is what is happening there."*

(IP 11)

Women described sometimes having to walk far or to unsafe places for work or in search of basic needs which was often associated with exposure to sexual violence. Mothers also described having to send their children in search of necessities at times, and the children being at risk of gender-based violence when doing so.

*"But it was one time back when I was just moving around, selling the clothes I made. . .it was late and at around seven PM I met five guys on the way when I was just going back home. So, I was just down in the valley, it's where I met those guys. . .five of them. They raped me."*

(IP12)

*Engaging in transactional sex for survival.* In the face of the material and economic hard-ships experienced in the camp, women sometimes described resorting to survival sex as a means of coping whereby they agreed to engage in sex in exchange for basic needs. Some also had daughters who engaged in more regular transactional sex to meet their fundamental needs and ensure their survival.

*"If I spend a whole day hungry, and I sleep hungry. . .and a man comes and offers me 10 000 shillings if I sleep with him for my children to get food. A man who is not minding about the family, has another wife. You will accept the offer so you can go and buy food for the children. (. . .) It happened to me. I cannot deceive."*

(IP 5).

**Striving for economic independency.** In their efforts to provide, women sought new ways to secure basic needs for themselves and their families, which in turn challenged previ-ously established gender roles. In the household, however, the gender norms were preserved through violence, and women were denied resources which were still in control of the hus-band, thus leaving the women with no option but to continue striving for economic independence.

*The unintended effect of the WFP money.* The United Nations World Food Program (WFP) provides refugees with monthly relief cash in the settlement. Many women described that the husbands used this money for alcohol and other purposes, leaving nothing for the women and their families.

*"Another risk factor (for GBV) in the camp is the food, the cash assistance that we get from the World Food Program. You find the man going there, picking up the money, and instead of bringing it home, he takes it to the bar. When he comes back and the wife asks, where is the money that you went to pick up, there will be war."*

(IP 8)

In order to receive the money, one had to provide an attestation form where families and spouses were registered together. The registered recipient of the relief cash could be either the

man or the woman in the household. Even if the women received the money, husbands may use violence to seize it.

> *"The man comes with the money. . .Or the woman gets the money (from the World Food Program), then after getting the money the man will see it and try all possible ways of beating her to get that money to go and drink alcohol. Then after drinking alcohol the children will stay hungry, and they will be looking at the mum. . ."*

(IP 11)

Furthermore, as the money was oftentimes denied the women and their families by the husbands, being registered together with the husband was often considered a problem as the money and other support was then distributed to them as a family.

> *"Sometimes if you are on the same attestation form as your husband. . .When you report a case of gender-based violence and they are seeing you are on the same attestation as your husband they just tell you"you are on the same attestation with your husband, you go and manage your issues". And every other kind of help that people seek they always tell women they have husbands. They mostly listen to that one who is not on the same attestation with the husband, that one they can listen to."*

(IP 3)

*Being under the economic control of the husband.* Denying resources was a frequently described form of intimate partner violence. Money was often used by the men for their own purposes, leaving the women and their families with nothing. If the woman happened to have been in possession of the money or questioned the husband's use of it, she might be subjected to physical violence. There were also situations described where the husband left and took all possessions with him or when women were denied shelter by being forced outside of the house. Women also described being denied food which they had earned and prepared themselves.

> *"After working, bringing home that food, you cook. . .and still you might not even eat that food because the husband will start you a fight. . .and not allow you to eat that food."*

(IP 2)

Participants described being beaten or emotionally abused when asking for the resources they were denied. Some were also punished by being denied other resources such as shelter.

> *"If you ask the husband about that money or food, the man will start beating you and say"-how dare you ask me for money?". Sometimes they chase women out of the house, and you must sleep outside because you asked for the money. . ."*

(IP 2).

*IPV as power and gender roles enforced through violence.* Participants described intimate partner violence in the form of physical, emotional, and sexual abuse. Some abusive situations arose from refusing to obey the husband, from talking back, and not giving him enough attention. Wanting empowerment or asking the husband for money he does not have were also mentioned as risks for domestic violence.

*"When you refuse to abide by what he wants, he will start beating you and also will confiscate the household attestation."*

(IP 5).

Although a few women described IPV in both conflict and the refugee settlement, most cases of IPV described had happened in the camp, perpetrated by husbands they came with or by husbands they met in the settlement. In the case of one respondent, the husband first started being violent after they had been attacked by rebels in DRC.

*"In Congo. . .they carried us from the house, they pushed us into the bush. Then after the husband also started to disrespect me, beating me, abusing me. . .then we came to Uganda, running from the war. Reaching Uganda, the man is doing the same, beating me. I am sleeping outside; the man chases me out of the house. Those things."*

(IP 11)

Intimate partner violence was also described in situations where women diverged from traditional gender expectations such as having difficulties conceiving. It was described that husbands became abusive or left when they found out, or when rumors were spread that the woman could not conceive.

*"Then the other man is also asking for a child, and I am not giving birth. Now the man sleeps in the bar. He comes and beats me, tells me abusive words. He is not contributing with anything in the house. . .he just comes home late at night, wants food or whatever else. The man is not providing, (he is) beating, doing things. . .that is what I am going through."*

(IP 10).

*Financial support and decreasing economic dependency the only way*. Increased financial support was mentioned as key to improving living conditions and reducing gender-based violence. Increased cash assistance could make one less dependent and less prone to being exposed to gender-based violence when pursuing basic needs.

*"If that food or cash assistance is increased, gender-based violence would be reduced because here gender-based violence is mostly caused by poverty. So, I think if they increase (the cash assistance) maybe gender-based violence would be reduced in the camp."*

(IP7)

It was also emphasized that it was important for the women to have control of the attestation money, that is the monthly ration that they received from the World Food Bank. One way that was mentioned was the need to be registered separately from the husband, as households were usually registered together and thus got money handed out to them as a unit.

*"With 12000 a month you are not able to serve your family. And if they check on your attestation and they find that you have a husband, no other (financial) support will be given because they know you have a husband, so you are able to cater for your family. You will not get any help apart from that 12000 and that is what is making life more difficult."*

(IP2)

The participants mentioned finding ways of their own to take control of the monthly ration by confiscating, hiding, giving the money to businessmen for food or just hurrying to pick up the money so that the husbands would not use the money for themselves.

*"The attestation, I keep it at the neighbor's place, because he (the husband) can confiscate it and he takes it for his personal use mostly."*

(IP5)

Participants emphasized the need for further skills and vocational training to be able to open up small businesses or earn a living, and they believed that this would improve not only the overall situation and poverty but also dependency on the husband and exposure to gender-based violence outside of the house.

*"What can be done for women and girls is give them something to do. Like the IRC program of supporting women with life skills. Yes, they have started it, but the service is still raw. Maybe if they can learn a skill, they have something to do outside, like a startup. If they give them different skills or some support, they will be able to do some business. Those who have skill, could get support to use their skills so that they can be earning in their home."*

(IP1)

**Building resilience.** Being faced with all the challenges brought about by conflict, migration, and poverty, the participants tried to navigate and build a new reality in the refugee camp. Participants showed great resourcefulness, strength, and creativity as they sought ways to tend to their families' needs, deal with adversities and rebuild their lives and networks.

*Building new support systems.* Many women described creating friendships and coping by meeting fellow women. Women described going to their friends and neighbors with difficulties and challenges such as GBV in order to gain emotional support, advice, and sometimes practical support. They described how neighbors and other community members had advised them and motivated them to get help. Participants also described informing and being informed by others of available help in the settlement.

*"Another thing that we do is to meet with other fellow women and girls. We meet and share the challenges that we are going through and sometimes we find solutions to some of the challenges."*

(IP 6)

*"My whole family is in Congo and the others died. Here what I have are my neighbors. When I get a problem, I turn to my neighbor. I explain my challenge. If you don't know Alight he will direct you to Alight. . .tell you who is a partner, who deals with problems of children and adults. Because there are times when you decide you do things on your own and you spoil everything."*

(IP 5)

*Avoiding violence and seeking a safe space.* Women described avoiding violence by avoiding places where they have been exposed to GBV before, including their homes, by for instance spending time outside of the house and engaging in different activities. Another participant

was employing strategies to avoid provoking violence, such as by staying quiet when at home. Some women even described leaving the house after abuse from their husbands.

*"I manage controlling GBV by like moving, joining life skill activities here, here, here, here and here. Then after I will go and make mandazi, and those baskets. . .I know how to make baskets, but I don't have material to do it. And I also make chapatis, then time will pass. . .and after time passing, I will go back and sleep and keep quiet."*

(IP 10)

Participants turned to NGO safe spaces for women and girls where they expressed feeling comfortable and supported. In these spaces they felt they could share their experiences as well as build new networks. Sharing their experiences in these spaces was described as positive for their emotional well-being and gave a sense of relief. A participant also mentioned warning each other of dangers such as risks of sexual violence in certain working locations.

*"IRC provides a safe space for women and girls where they come and share experiences. . .In Congo we had no time to sit in a group to have a session, or discussion. . .no, but here in the women's center for girls and women they come, they share. . . You find someone is stressed and then when you go out from here, you find that there is less stress because you've been speaking to one another through our sessions."*

(IP10)

*Needing to spread awareness*. Women described a duality of awareness, where awareness and information about GBV was spread and talked about openly in the community while at the same time people were unwilling to change and lacked an understanding of rights and violence. Furthermore, women perceived a gendered awareness gap and felt that there should be more efforts to inform and spread awareness to men and that women's efforts to spread information to men were met with resistance.

*"Us women we always come and get that information. But us reaching them, if we start giving that information to our partners they will start saying "eh, you went there to learn how to misbehave with me. If it is like that then just go and remain there or don't go there again". Then you will find it is these men that need to be told much more than women, because if these men come to know that treating a woman like this is bad, they will stop doing it."*

(IP2)

## Discussion

To the best of our knowledge this is one of few qualitative studies to investigate women's perspectives and experiences of GBV in a refugee camp in Uganda. The findings paint a picture of complex processes involved in forced migration and life in a refugee camp setting that in turn created situations of vulnerability to GBV. Women described fleeing the violence experienced in conflict and finding themselves navigating in a new reality in the refugee camp with new structural and individual challenges including increased marginalization, poverty and continued exposure to GBV, illustrated by the overarching theme of "no easy escape route from gender-based violence". The risks and level of gender-based violence were high throughout the entire process, in conflict and in displacement. The majority of the interviewed women were

refugees from the Democratic Republic of Congo (DRC), from a prolonged conflict known to have a high prevalence of sexual and physical violence against women and girls [39]. Almost all participants described extensive sexual and physical violence by the rebels in DRC. In the refugee camp levels of gender-based violence remained high but the perpetrators, risk contexts, and drivers of GBV somewhat shifted. Previous studies from Northern Uganda, East Timor, and refugee settlements in Rwanda and Iran have all shown that IPV prevalence is high in conflict affected populations [26, 40–42]. In our study most cases of IPV described had occurred in the refugee camp in the host country. Similar observations were made in a qualitative study of refugees living in urban and camp settings in Ethiopia where IPV was mostly reported to transpire in the host country [28]. Another study from East Timor showed that some IPV levels among refugees remained the same or increased in the post-conflict period [42]. Several factors such as disrupted support systems, lack of essential resources and marginalization were some of the consequences of migration described by the participants, all of which are known risk factors for increased GBV in humanitarian settings [6, 19, 20].

Previous studies from similar settings such as refugee settlements in Kenya and Northern Uganda suggest that there is often a shift in power and gender roles that occurs within the household in displacement whereby it is not uncommon for female refugees in camps to become the primary providers for the family and the husbands to lose economic and social status [43, 44]. Such findings are very much in line with those in our study where women reported becoming the sole providers of the household, either by losing a spouse in conflict or because the husbands seemed to either lose motivation or ability to provide for the family. The women expressed being left alone in their role as a parent on a practical and emotional level. This, together with the amplified lack of basic needs and work opportunities, resulted in women seeking various ways to provide, often risking being exposed to physical and sexual violence. A common way to pursue an income was looking for casual labor opportunities in the host community or soldier barracks where women described frequent use of sexual violence against them. Another risk context for sexual and gender-based violence (SGBV) by strangers was walking long distances in search for basic necessities or work. The use of transactional sex for money or basic needs was also described. Similar risk contexts for SGBV as well as the use of transactional sex as a survival strategy have been seen in a qualitive study from another refugee camp in northwestern Uganda as well as in a qualitative study on internally displaced populations in Haiti [45, 46]. Women described a cumulative effect of the constant survival stress and violence resulting in a feeling of exhaustion and fatigue.

At the same time as the economic roles were shifting, domestic violence seemed to increase. Women continued to find themselves under the economic control of the husband who continuously denied them resources and money. This is in line with figures from UNHCR showing that denial of resources is a common form of IPV reported among refugees in settlements in Uganda [25]. Our results seem to point to an unintended effect of the monthly relief cash provided by the World Food Program (WFP) for food and necessities. According to the women, this money was often spent by the husband on alcohol or prostitution, leaving little for the family. Asking for the money created tension and violence in the household, similar to findings in a qualitative study from a refugee camp in Kenya where a common trigger of violence was asking the husband for resources [43]. Even if the women picked up the money, the husbands were described as using violence to obtain it. Other forms of IPV including physical and emotional violence were also frequently described, as a result of refusing to obey the husband, asking for resources, or as punishment for not living up to gender norms such as the ability to conceive. In these cases, violence serves as a way to assert power and gender roles. As in previous qualitative studies from internally displaced communities in Colombia and refugee camps in Ethiopia, South Sudan, Kenya, and Iraq, the shifting economic power and gender roles

described above are thought to trigger an increase in domestic violence [20, 24, 28], where the male in the household attempts to regain power and validation of his masculinity through violence [24, 43]. Among other factors believed to increase risk of IPV in conflict-affected populations are alcohol and substance abuse [20, 26], something that seems to emerge in our data as well. Two cross-sectional studies, one from the occupied Palestinian territory and another one from community health centers in Boston, have shown an association between men's previous exposure to political, war-related violence and IPV [47, 48]. Also, a study from two Congolese refugee camps in Rwanda showed a significant correlation between a woman's history of outsider violence and IPV [40], an association that might very well be at play in this study setting.

Since the resources and power in the household were maintained by the husband, the only way to improve the situation and decrease poverty and gender-based violence outside and inside of the household was to decrease the women's dependency on the husband by separating the attestation, increasing the monthly ration, and increasing skills training. The importance of increasing the financial support was emphasized after reductions in the monthly rations in recent years following the COVID-19 pandemic, which had worsened the already dire circumstances [49].

Our findings show many ways in which women attempted to rebuild their lives in the camp to manage the rampant poverty and violence. Building new support systems and relationships with people in the community and finding ways to avoid violence were some of the methods mentioned. The NGO safe spaces for women and girls, where most of the women were interviewed, were described as spaces where they could build networks, share experiences, get support and be safe from violence. Previous qualitative studies from Spain and asylum-centers in Belgium have described the importance of building new networks and social support systems, emphasizing their role as an important psychosocial resource of resilience and coping in women who have experience of GBV [50, 51]. Avoiding violence by for instance evading their partners temporarily or finding strategies to minimize risk of violence from their partners has been described as a method of covert resistance applied by women exposed to GBV [52]. These methods are especially useful in settings like these where opportunities for other forms of resistance and risk mitigation are scarce.

Our findings suggest an interesting duality of awareness around GBV. On the one hand there seems to be awareness and open conversations about issues of GBV in the community. On the other hand, women describe a lack of understanding and a lack of willingness to change and apply this knowledge in the community. Especially men are said to resist when women attempt to spread information to partners. It is therefore emphasized that more awareness campaigns be directed at men. Without addressing the perpetrators of violence, little is believed to change [53].

## Methodological considerations

The study had both strengths and limitations. One limitation was having to use interpreters which affected the communication and rapport between the participants and interviewer. Furthermore, the interpreters were local community activists and not professional interpreters which could potentially affect the quality of translation. However, the interpreters were all trained in gender-based violence case management and were familiar with the topics and language at hand as well as with local expressions. They were known to the community and worked at the organizations dealing with GBV and thus were known to the participants. This brought about a sense of familiarity and was thought to increase a feeling of rapport and safety, especially as the interviewer was someone unfamiliar from the outside. This in turn increased the likelihood of obtaining more comprehensive data and enhanced credibility. Furthermore,

the interpreters had in many instances already been involved in these cases before. Their presence and familiarity could, however, potentially make the participants less at ease to share personal and sensitive information, which is why confidentiality forms were always signed by the interpreters in front of the participants.

The interviewer had an entirely different background and was unknown to the participants, which could influence the rapport, interaction and quality of the interviews and analysis. The interviewer being someone from the outside could, however, also strengthen the dialogues as the participants might have felt more at ease sharing sensitive information with someone with whom they had no prior relationship. Furthermore, the interviewer has worked in health care with vulnerable patients for many years and had undergone training in consultation techniques, which may have enhanced the depth and sensitivity of the interview. In order to increase a sense of familiarity the main researcher presented herself, including relevant experiences and motivation behind the study prior to starting the interviews. To reduce bias and influence of preconceptions and thus increase both credibility and confirmability of the results, the interview questions were revised according to feedback from experts and representatives of the Office of the Prime Minister Uganda. Also, the primary researcher engaged in a reflexive process with reflexive notes and peer debriefing with co-authors and experts including local community activists and social workers, as well as a local research assistant working with issues of gender-based violence at the Ministry of Gender Labour and Social Development in Uganda. However, to further strengthen confirmability, member checking could have been employed.

To better capture variations in perceptions and experiences, the participants were recruited by trained GBV service providers working with several different partner organizations mentioned above. For ethical reasons, however, only women over eighteen years of age were included in the study. As there is reason to believe GBV affects individuals across the younger age groups as well, a possible limitation is the lack of younger participants. Furthermore, due to the demographic of the refugee camp almost all participants were refugees from the DRC. Refugee settlements may vary depending on national policies and resources, whether the settlements are protracted or short-term emergency settlements, in rural or urban settings, and whether they are government or internationally managed. The infrastructure and size of the settlements, the services in place as well as the populations and their needs may vary and affect the transferability of the results. However, considering that 75% of the world's refugees and other people in need of international protection are hosted by low- and middle income countries such as Uganda [54], this setting is believed to share many structural and social similarities with many other refugee settlements in terms of available resources, infrastructure, and vulnerability of the populations, thus, enhancing the transferability of these findings.

The number of participants in the study was appraised using the concept of information power as described by Malterud et al, whereby information power depends on factors such as the study aim, the quality of dialogue, the specificity of the sample, use of established theory and analysis strategy [55]. The number of participants included was deemed to be appropriate in light of these aspects.

Although the consent forms were thoroughly described and signed by the participants, one cannot ignore the perceived power imbalance in these contexts which needs to be taken into account in understanding the data. Another possible limitation was the presence at times of too many parties during the interviews. The participants were always asked if they were comfortable and consented to present parties, and the additional parties were either another community activist or counselor known to the participant there to provide support in case of distress after the interview.

Finally, the findings and data interpretation were discussed with co-authors throughout the process until consensus was achieved, thus increasing the credibility and confirmability of the results.

## Conclusion

Our study shows the complexity of the widespread problem of gender-based violence in humanitarian settings such as refugee camps. Migration and life in refugee camps are associated with many structural and personal changes affecting risks of gender-based violence both outside and inside of the household. Destabilized conditions and poverty increase women's risk of GBV outside of the household and destabilized power and economic gender roles seem to increase risk of violence within the household.

Comprehensive interventions across various dimensions are therefore needed to address GBV. These include efforts towards improving access to and safe alternatives for securing basic needs for women, enhancing their control of these resources within the household, and addressing power relations and gender norms within the household, community, as well as between the host and migrant communities. Additionally, further efforts are needed to support the coping mechanisms described by the participants, such as community and network building.

Further research is essential to better understand GBV in refugee camp settings, as well as the perception and effectiveness of the existing interventions and services. Including male participants in future studies could also provide a deeper understanding of the mechanisms at play in this context. A stronger commitment globally and locally to address the risks of GBV in conflict, migration and displacement is needed especially as the number of conflicts and refugees seems to be increasing world-wide.

## Acknowledgments

We would like to thank all the participants for sharing their experiences, invaluable thoughts and perspectives. We would also like to thank the partner organizations, translators, local community activists and social workers for their important work and assistance during and after the interviews.

We would like to express our appreciation to Lorna Kigozi for her invaluable help with data collection and field notes. We are also grateful to Dr Angela Nakafeero, Commissioner Gender and Women Affairs at the Ministry of Gender, Labour and Social Development and the Office of the Prime Minister in Uganda for granting us access and permission to conduct the study in the Kyaka II refugee camp as well as giving much valued feedback on the interview guide.

## Author Contributions

**Conceptualization:** Miriam Lukasiak, Gilbert Tumwine, Anette Agardh.

**Data curation:** Miriam Lukasiak.

**Formal analysis:** Miriam Lukasiak, Jack Palmieri, Pia Svensson, Anette Agardh.

**Funding acquisition:** Anette Agardh.

**Investigation:** Miriam Lukasiak.

**Methodology:** Miriam Lukasiak, Jack Palmieri, Pia Svensson, Gilbert Tumwine, Anette Agardh.

**Resources:** Anette Agardh.

**Supervision:** Jack Palmieri, Gilbert Tumwine, Anette Agardh.

**Writing – original draft:** Miriam Lukasiak.

**Writing – review & editing:** Miriam Lukasiak, Jack Palmieri, Pia Svensson, Gilbert Tumwine, Anette Agardh.

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
