## [Decision Letter · Decision Letter 0]

30 Aug 2024

PONE-D-24-21429

Exploring perceptions and experiences of gender-based violence among women in a refugee camp setting in Uganda - a qualitative study

PLOS ONE

Dear Dr.  Lukasiak,

Thank you for submitting your manuscript to PLOS ONE. After careful consideration, we feel that it has merit but does not fully meet PLOS ONE’s publication criteria as it currently stands. Therefore, we invite you to submit a revised version of the manuscript that addresses the points raised during the review process.

Editor comments

Do you think the finding, and recommendations are inline? Please make sure all concerns under methodology are incorporated appropriately based on the suggestions provided by reviewers.

What makes unique your study from study conducted three years back in Uganda among refugee population? http://dx.doi.org/10.1353/eas.2018.0010

Please submit your revised manuscript by Oct 14 2024 11:59PM. If you will need more time than this to complete your revisions, please reply to this message or contact the journal office at plosone@plos.org. Please include the following items when submitting your revised manuscript:

We look forward to receiving your revised manuscript.

Kind regards,

Derebe Madoro Bunte

Academic Editor

PLOS ONE

Journal Requirements:

2. For studies involving third-party data, we encourage authors to share any data specific to their analyses that they can legally distribute. PLOS recognizes, however, that authors may be using third-party data they do not have the rights to share. When third-party data cannot be publicly shared, authors must provide all information necessary for interested researchers to apply to gain access to the data. (https://journals.plos.org/plosone/s/data-availability#loc-acceptable-data-access-restrictions) 

Additional Editor Comments:

Do you think the finding, and recommendations are inline? Please make sure all concerns under methodology are incorporated appropriately based on the suggestions provided by reviewers.

What makes unique your study from study conducted three years back in Uganda among refugee population? http://dx.doi.org/10.1353/eas.2018.0010

Reviewer comments to author

1. Is the manuscript technically sound, and do the data support the conclusions?

Reviewer #1. Partly

Reviewer #2. Yes

Reviewer #3. Yes

Reviewer #4. Yes

2. Has the statistical analysis been performed appropriately and rigorous?

Reviewer #1. Yes

Reviewer #2. Yes

Reviewer #3. N/A

Reviewer#4. N/A

3. Have the authors made all data underlying the findings in their manuscript fully available?

Reviewer #1. No

Reviewer #2. Yes

Reviewer #3. No

Reviewer#4.Yes

4. Is the manuscript presented in an intelligible fashion and written in standard English?

Reviewer #1. Yes

Reviewer #2. Yes

Reviewer #3. Yes

Reviewer#4.Yes

5. Review Comments to the Author

Please use the space provided to explain your answers to the questions above. You may also include additional comments for the author, including concerns about dual publication, research ethics, or publication ethics. (Please upload your review as an attachment if it exceeds 20,000 characters).

Reviewer#1.

Dear authors! You are doing a great job. Thank you for conducting this study. Generally, your manuscript has no line number to comment on line by line. So, please try to give them line numbers. There are redundancies in the ideas and words thought out in this manuscript. Please try to reduce repetition of the same ideas and words and rewrite them in a clear and brief manner.

Ethical consideration section

This study ignored the right of study participants' spouses or partners to let their spouses or partners to participate in this study. Similarly ethical concerns pertaining to refugee camps were not addressed. Who were the data collectors? Were they females or males? Because males are not appropriate for this sensitive issue of data collection.

Recommendation section

Your recommendations not in line with your study findings. The recommendation forwarded, “Interventions across various dimensions are therefore warranted to address GBV and gendered power dynamics on multiple levels,” was not specific; it seems general and vague. Please rewrite a specific recommendation based on your study findings.

Abstract: - It contains all the scientific content and is written well. But, your recommendation was not appropriate.

Introduction correct as “Background”

Background: - it was well written and appropriately showed the problem, why this study was important and the gaps seen from previous studies. However, it is lengthy, as if it will be revised and rewritten brief manner. It is better to search for and incorporate previous studies reports about women’s GBV perceptions and experiences into this study background.

Has editorial errors for instances; - 89, 4 26,4 1,5 ??

Methods: can be correct as "Methods and materials."

For this qualitative study, study setting, design, period, source of population, sampling technique, data collection tools and procedures, and ethical consideration were not separately and clearly written. So, it is better if all the above subtopics under Methods and materials are rewritten in separate subsection and in clear manner.

It was not clear how the study participants were selected and invited from women who were supported by different organizations within single refugee camp

Under the subtopic “Interview Procedure and Consent” you stated that” Eligible participants of the interviews were informed of all aspects of the study by the principal investigator; its purpose, process, risks and benefits, voluntary participation, confidentiality, and all safety and security precautions The participants were informed of their right to withdraw from the study at any time without any explanation or consequences. .” This issue is to address ethical consideration, so it is better to take it to the ethical consideration section.

Results: appropriate and well done. But there are sections more focused on socio-economic and health problems that study participants faced at refugee camps. These were not in line with the study title, “Study participants’ perceptions and experiences of GBV”. Your study results should to stick with your study title

Discussion

In my opinion, it was well done and scientifically appropriate.

Strengths and limitations

This study limitation was not well addressed, for instance data collection methods such as focused group discussion and males were not participated to explore their thoughts and perceptions regarding GBV

Reviewer #2.

Yes I would like to begin by expressing my appreciation to the authors for their insightful exploration of gender-based violence among women in a refugee camp setting in Uganda. The article sheds light on a critical issue that affects countless individuals worldwide, and the authors' dedication to addressing this topic is commendable. Through their thorough research and thoughtful analysis, they have provided a valuable contribution to the ongoing conversation on gender-based violence, highlighting both the characteristics of GBV experienced both in conflict and displacement and the perpetrators as well as the perceived risk contexts.

My concerns on the article are mainly focused on the methodology, as written below.

1. Type of Qualitative Study

Does this study fall into any specific type qualitative study, such as phenomenology, grounded theory?

2. Reflexivity

What is the background of the researchers and interviewer? Credentials, occupation, gender, training and experience? What are the relationships between the researcher and the participants? What reflexivity has been done to ensure transparency and enhance validity of the research. There are no specific information on interviewer characteristics.

Reviewer #3. I feel the study is well-executed and provides significant insights into the experiences of gender-based violence (GBV) among women in a refugee camp in Uganda. The qualitative approach is suitable for exploring such a sensitive and complex issue, and the findings contribute meaningfully to the existing literature on GBV in humanitarian settings.

Also need some improvements

The methods section could benefit from a more detailed description of the content analysis process, including how themes were identified and categorized. It would also be helpful to know if any software tools were used for the analysis.

While the results are well-organized, additional direct quotes from the interviews could enhance the richness of the qualitative data.

The discussion section could provide a more explicit discussion of the limitations of the study, particularly regarding the generalizability of the findings.

The data availability statement should be clarified to ensure that the underlying data is accessible, or a clear explanation should be provided if there are restrictions due to ethical or privacy concerns.

3. Triangulation

What kind of triangulation has been conducted in the data?

4. Number of participants

What method was used to ensure the data is saturated?

5. Data collection

Apart from the transcripts, were there any other types of data that were recorded and analyzed, such as field notes, observations, photos, documents?

6. Validity

What measures have been taken to enhance validity of the qualitative research? Member checking, participant checking?

I suggest the article should follow the Consolidated criteria for reporting qualitative research (COREQ).

Reviewer#4. The manuscript presents a qualitative study that explores the perceptions and experiences of gender-based violence (GBV) among women residing in a refugee camp in Uganda. The study's primary objective is to gain a comprehensive understanding of the nuanced manifestations of GBV within this vulnerable population. It seeks to examine the cultural and environmental factors that contribute to these experiences. Through in-depth interviews and focus group discussions with the women in the camp, the authors aim to shed light on the challenges these individuals face, their coping mechanisms and available support structures.

This study addresses a critical and relatively understudied aspect of gender-based violence in refugee settings. It provides valuable insights that not only contribute to academic discourse but also have direct implications for practical interventions. The qualitative approach employed in this research is particularly well-suited for exploring the intricate and sensitive issues surrounding GBV in this context. The manuscript demonstrates a well-organized structure, featuring a clear introduction that highlights the significance of the study, a detailed description of the methods employed, and a discussion that situates the findings within the broader literature.

However, there are some areas where the manuscript could be strengthened to improve clarity, rigour, and overall impact.

While the authors offer recommendations for practice and policy, there is room for further development. Specifically, the manuscript would benefit from a clearer explanation of how the findings can inform specific interventions or policy changes to address GBV in refugee settings.

Conclusion:

The conclusion effectively summarizes the main findings, but it could be enhanced by restating the study's contribution to the field and proposing directions for future research.

6. PLOS authors have the option to publish the peer review history of their article (what does this mean?). If published, this will include your full peer review and any attached files.

Do you want your identity to be public for this peer review? For information about this choice, including consent withdrawal, please see our Privacy Policy.

Reviewer #1. No

Reviewer #2. No

Reviewer #3. No

Reviewer#4.No

Reviewers' comments:

Reviewer's Responses to Questions

**Comments to the Author**

1. Is the manuscript technically sound, and do the data support the conclusions?

Reviewer #1: Partly

Reviewer #2: Yes

Reviewer #3: Yes

Reviewer #4: Yes

2. Has the statistical analysis been performed appropriately and rigorously? 

Reviewer #1: Yes

Reviewer #2: N/A

Reviewer #3: Yes

Reviewer #4: N/A

3. Have the authors made all data underlying the findings in their manuscript fully available?

Reviewer #1: No

Reviewer #2: No

Reviewer #3: Yes

Reviewer #4: Yes

4. Is the manuscript presented in an intelligible fashion and written in standard English?

Reviewer #1: Yes

Reviewer #2: Yes

Reviewer #3: Yes

Reviewer #4: Yes

5. Review Comments to the Author

Reviewer #1: Dear authors! You are doing a great job. Thank you for conducting this study. Generally, your manuscript has no line number to comment on line by line. So, please try to give them line numbers. There are redundancies in the ideas and words thought out in this manuscript. Please try to reduce repetition of the same ideas and words and rewrite them in a clear and brief manner.

Reviewer #2: i feel the study is well-executed and provides significant insights into the experiences of gender-based violence (GBV) among women in a refugee camp in Uganda. The qualitative approach is suitable for exploring such a sensitive and complex issue, and the findings contribute meaningfully to the existing literature on GBV in humanitarian settings.

Also need some improvements

The methods section could benefit from a more detailed description of the content analysis process, including how themes were identified and categorized. It would also be helpful to know if any software tools were used for the analysis.

While the results are well-organized, additional direct quotes from the interviews could enhance the richness of the qualitative data.

The discussion section could provide a more explicit discussion of the limitations of the study, particularly regarding the generalizability of the findings.

The data availability statement should be clarified to ensure that the underlying data is accessible, or a clear explanation should be provided if there are restrictions due to ethical or privacy concerns.

Reviewer #3: I would like to begin by expressing my appreciation to the authors for their insightful exploration of gender-based violence among women in a refugee camp setting in Uganda. The article sheds light on a critical issue that affects countless individuals worldwide, and the authors' dedication to addressing this topic is commendable. Through their thorough research and thoughtful analysis, they have provided a valuable contribution to the ongoing conversation on gender-based violence, highlighting both the characteristics of GBV experienced both in conflict and displacement and the perpetrators as well as the perceived risk contexts.

My concerns on the article are mainly focused on the methodology, as written below.

1. Type of Qualitative Study

Does this study fall into any specific type qualitative study, such as phenomenology, grounded theory?

2. Reflexivity

What is the background of the researchers and interviewer? Credentials, occupation, gender, training and experience? What are the relationships between the researcher and the participants? What reflexivity has been done to ensure transparency and enhance validity of the research. There are no specific information on interviewer characteristics.

3. Triangulation

What kind of triangulation has been conducted in the data?

4. Number of participants

What method was used to ensure the data is saturated?

5. Data collection

Apart from the transcripts, were there any other types of data that were recorded and analyzed, such as field notes, observations, photos, documents?

6. Validity

What measures have been taken to enhance validity of the qualitative research? Member checking, participant checking?

I suggest the article should follow the Consolidated criteria for reporting qualitative research (COREQ).

Reviewer #4: The manuscript presents a qualitative study that explores the perceptions and experiences of gender-based violence (GBV) among women residing in a refugee camp in Uganda. The study's primary objective is to gain a comprehensive understanding of the nuanced manifestations of GBV within this vulnerable population. It seeks to examine the cultural and environmental factors that contribute to these experiences. Through in-depth interviews and focus group discussions with the women in the camp, the authors aim to shed light on the challenges these individuals face, their coping mechanisms and available support structures.

This study addresses a critical and relatively understudied aspect of gender-based violence in refugee settings. It provides valuable insights that not only contribute to academic discourse but also have direct implications for practical interventions. The qualitative approach employed in this research is particularly well-suited for exploring the intricate and sensitive issues surrounding GBV in this context. The manuscript demonstrates a well-organized structure, featuring a clear introduction that highlights the significance of the study, a detailed description of the methods employed, and a discussion that situates the findings within the broader literature.

However, there are some areas where the manuscript could be strengthened to improve clarity, rigour, and overall impact.

While the authors offer recommendations for practice and policy, there is room for further development. Specifically, the manuscript would benefit from a clearer explanation of how the findings can inform specific interventions or policy changes to address GBV in refugee settings.

Conclusion:

The conclusion effectively summarizes the main findings, but it could be enhanced by restating the study's contribution to the field and proposing directions for future research.

6. PLOS authors have the option to publish the peer review history of their article (what does this mean?). If published, this will include your full peer review and any attached files.

Reviewer #1: No

Reviewer #2: No

Reviewer #3: No

Reviewer #4: No

---

## [Author Response · Author response to Decision Letter 0]

20 Sep 2024

Dear Editor

We would like to thank you and the reviewers for the valuable feedback. We have revised the manuscript in line with this feedback. Please see the revised version attached, and our responses to the reviewers’ comments, point by point, listed below. In the revised version of the manuscript additional sources have been added to the reference list (#29, #37, #38, #54, #55).

With kind regards,

Miriam Lukasiak

Comments from Editor

1. Do you think the finding, and recommendations are inline? Please make sure all concerns under methodology are incorporated appropriately based on the suggestions provided by reviewers.

Response

Yes, we believe that the revised recommendations are now well in line with the findings. We have paid due attention to the methodological concerns.

2. What makes unique your study from study conducted three years back in Uganda among refugee population? http://dx.doi.org/10.1353/ eas.2018.0010

Response

Dear Editor, thank you for bringing this study to our attention. We have now added a citation to this study in our manuscript. Although both studies illustrate the extensive exposure and complexity underlying gender-based violence in refugee camp settings in Uganda, they have slightly different focus, methods and findings and thus complement each other well. The former study has more focus on the institutional capacity and services in place to address gender-based violence, something that is not explored within the scope of our study. Our study on the other hand has more focus on the underlying processes driving different forms of gender-based violence, non-partner and intimate partner violence, respectively, a distinction that is not clearly formulated in the former study. The former study also describes ethnic disparities and specific cultural norms as drivers of gender-based violence, something that was not apparent in our results. The forms of violence described in the two studies were also slightly different, where their findings suggest that the most common forms of violence described were physical and sexual. In our study, however, emotional and social violence was described extensively and was closely intertwined with the underlying drivers of other forms of gender-based violence. An example of social violence was the denial of basic needs by intimate partners, exposing women to non-partner violence as women had to pursue basic needs. The coping mechanisms described in the two studies also differed from one another, where our study points more to community and network building as well as the construction of safe spaces as coping mechanisms. In conclusion, we believe the studies have similarities and differences and complement each other well.

Comments from Reviewer #1

1. Dear authors! You are doing a great job. Thank you for conducting this study. Generally, your manuscript has no line number to comment on line by line. So, please try to give them line numbers.

Response

Thank you so much for your comment, line numbers are now included.

2. There are redundancies in the ideas and words thought out in this manuscript. Please try to reduce repetition of the same ideas and words and rewrite them in a clear and brief manner.

Response

Thank you. We have tried to eliminate repetition in the manuscript.

3. Ethical consideration section 

This study ignored the right of study participants' spouses or partners to let their spouses or partners to participate in this study.

Response

Thank you for your comment. Considering the nature of this sensitive topic that could potentially be related to partners, asking the participant’s spouse for permission to participate in a study on gender-based violence would be considered unethical and could pose a risk to the participant’s safety and privacy. Ethically we were obliged to preserve the participants full confidentiality, and as the participants were of adult age we considered them to have full autonomy over the decision to participate.

4. Similarly ethical concerns pertaining to refugee camps were not addressed. Who were the data collectors? Were they females or males? Because males are not appropriate for this sensitive issue of data collection.

Response 

Thank you for this important comment. There are many ethical challenges and concerns pertaining to data collection of this kind in refugee camps often characterized by the disruption of security, stability and increased vulnerability of women and girls. In order to address these concerns we tried to adhere to the recommendations from WHO guidance on researching sexual violence in emergency settings (https://emergencymanual.iom.int/sites/g/files/tmzbdl1956/files/2022-08/WHO Ethical and Safety recommendations for researching, documenting and monitoring sexual violence in emergencies.pdf). This has been clarified in the manuscript.

Lines 251-255

Collecting data of this sensitive nature in refugee camps presents ethical challenges particularly due to the lack of stability and security, as well as the heightened vulnerability faced by women and girls in these environments. To address these concerns, we followed the recommendations outlined in the WHO guidance on researching sexual violence in emergency settings [38].

The data collectors were all carefully selected. The interviews were conducted by the main author, a woman, and the field notes were conducted by a female research assistant employed at the Ministry of Gender Labour and Social Development in Uganda. The interpreters were all trained GBV service providers, familiar with the topic and context at hand, and they were all employed by one of the partner organizations. All apart from two interpreters were female, but both males were trained GBV service providers with experience of providing support and services to GBV survivors. We have added this information to the text.

Lines 204-226

“The first author (ML) was of female gender, medical doctor and researcher, and has previous experience of providing medical assistance to survivors of gender-based violence. All apart from two interpreters were female. Field notes were conducted by a female research assistant employed at the Ministry of Gender Labour and Social Development in Uganda. The two male interpreters were both trained GBV service providers with extensive experience in providing support and services to GBV survivors. It was always verified that the participant was comfortable with the parties present in the room.”

5. Recommendation section 

Your recommendations not in line with your study findings. The recommendation forwarded, “Interventions across various dimensions are therefore warranted to address GBV and gendered power dynamics on multiple levels,” was not specific; it seems general and vague. Please rewrite a specific recommendation based on your study findings.

Response

Thank you for your insightful comment. The recommendations have now been revised.

Lines 854-867

“Comprehensive interventions across various dimensions are therefore needed to address GBV. These include efforts towards improving access to and safe alternatives for securing basic needs for women, enhancing their control of these resources within the household, and addressing power relations and gender norms within the household, community, as well as between the host and migrant communities. Additionally, further efforts are needed to support the coping mechanisms described by the participants, such as community and network building.

Further research is essential to better understand GBV in refugee camp settings, as well as the perception and effectiveness of the existing interventions and services. Including male participants in future studies could also provide a deeper understanding of the mechanisms at play in this context. A stronger commitment globally and locally to address the risks of GBV in conflict, migration and displacement is needed especially as the number of conflicts and refugees seems to be increasing world-wide. “

6. Abstract

 It contains all the scientific content and is written well. But, your recommendation was not appropriate.

Response

Thank you! The recommendation has been updated in the abstract.

Lines 27-33

“Our study showed the complexity of GBV in settings such as refugee camps, where various structural and individual changes involved in migration and life in a refugee camp seemed to create new risk contexts for GBV both inside and outside of the household. Interventions across various dimensions including addressing underlying conditions of marginalization and gendered power dynamics are therefore warranted to address GBV in refugee camps. Further research is essential to better understand this complex issue, as well as the perception and effectiveness of services and interventions in place.”

7. Background

Introduction correct as “Background”.

Response

Thank you, it has been corrected.

8. It was well written and appropriately showed the problem, why this study was important and the gaps seen from previous studies. However, it is lengthy, as if it will be revised and rewritten brief manner. It is better to search for and incorporate previous studies reports about women’s GBV perceptions and experiences into this study background.

Response

Thank you for your comment. The background has now been revised and updated to include more studies concerning women’s perceptions and experiences of GBV. 

Lines 109-113

“In a previous study on experiences of gender-based violence among refugee populations in Uganda, detection of and response to GBV were limited by unequal and gendered power relations and control of resources within the households. Due to feelings of dependency on their husbands, women often refrained from reporting or seeking support [29]. ”

As to the length of the background we have tried to delete some words in the text to make it briefer.

9. Has editorial errors for instances; - 89, 4 26,4 1,5 ??

Response

Thank you for pointing this out. The editorial errors have been corrected.

10. Methods: can be correct as "Methods and materials."

Response

Thank you, we prefer the word “Methods” and feel it is more appropriate for this qualitative study.

11. For this qualitative study, study setting, design, period, source of population, sampling technique, data collection tools and procedures, and ethical consideration were not separately and clearly written. So, it is better if all the above subtopics under Methods and materials are rewritten in separate subsection and in clear manner.

Response

Thank you for your input on the structure of this section. A separate subheading concerning ethical considerations has been added. 

Lines 250-265

“Ethical considerations

Collecting data of this sensitive nature in refugee camps presents ethical challenges particularly due to the lack of stability and security, as well as the heightened vulnerability faced by women and girls in these environments. To address these concerns, we followed the recommendations outlined in the WHO guidance on researching sexual violence in emergency settings [38]. Eligible interview participants were informed of all aspects of the study by the principal investigator; its purpose, process, risks and benefits, voluntary participation, confidentiality, and all safety and security precautions. The participants were informed of their right to withdraw from the study at any time without any explanation or consequences. It was also clarified that no services received would be affected should the participant choose to participate or not. No payment or incentives were provided for participants. The consent form was written in English and translated verbally to native language by translator. The participants then signed or left a thumb print. Ethical permission for the study was granted by the Research and Ethics Committee (REC), St. Francis Hospital, Nsambya, Uganda. Further approval to conduct the study at Kyaka II Refugee Settlement was given by the Office of the Prime Minister of Uganda.”

As pertains to the other subheadings we appreciate the clarity of the current subheadings and prefer not to have too many different levels.

12. It was not clear how the study participants were selected and invited from women who were supported by different organizations within single refugee camp.

Response

Thank you for your important point. We have now clarified the recruitment process.

Lines 163-167

”Participants were recruited face-to-face to avoid compromising their confidentiality and safety by written invitations or calls. They were recruited either when visiting facilities such as the 'safe spaces' for women and girls at partner organizations or selected by GBV service providers based on eligibility and exclusion criteria following prior contact with GBV support services.”

13. Under the subtopic “Interview Procedure and Consent” you stated that” Eligible participants of the interviews were informed of all aspects of the study by the principal investigator; its purpose, process, risks and benefits, voluntary participation, confidentiality, and all safety and security precautions The participants were informed of their right to withdraw from the study at any time without any explanation or consequences. .” This issue is to address ethical consideration, so it is better to take it to the ethical consideration section.

Response 

Thank you for your input, the section mentioned has now been moved to the section on ethical considerations, lines 250-265.

14. Results: appropriate and well done. But there are sections more focused on socio-economic and health problems that study participants faced at refugee camps. These were not in line with the study title, “Study participants’ perceptions and experiences of GBV”. Your study results should to stick with your study title.

Response

Thank you for your valuable input. Although we appreciate your point of view, these aspects were closely related to the perceptions and experiences of GBV among the women in the refugee camp and are an intrinsic part of the mechanisms underlying gender-based violence in the settlement. Being denied economic resources was moreover a type of intimate partner violence which was commonly experienced. Describing the structural and individual changes involved in migration and the new life in the refugee camp provides a more comprehensive understanding of the exposure to GBV in this context.

15. Discussion

In my opinion, it was well done and scientifically appropriate.

Response

Thank you.

16. Strengths and limitations

This study limitation was not well addressed, for instance data collection methods such as focused group discussion and males were not participated to explore their thoughts and perceptions regarding GBV.

Response

Thank you for your input, which we value highly. However, our perspective on the methodology varies slightly. Focus group discussions are useful for exploring attitudes and norms but are not as appropriate for studying perceptions and experiences, especially when the topic is highly sensitive, as was the case in this study. Including males in this study was not considered suitable for the focus of this study and would be a different study altogether. Therefore, we do not fell it relevant to mention these points as potential study limitations.

Comments from Reviewer #2

1. Yes I would like to begin by expressing my appreciation to the authors for their insightful exploration of gender-based violence among women in a refugee camp setting in Uganda. The article sheds light on a critical issue that affects countless individuals worldwide, and the authors' dedication to addressing this topic is commendable. Through their thorough research and thoughtful analysis, they have provided a valuable contribution to the ongoing conversation on gender-based violence, highlighting both the characteristics of GBV experienced both in conflict and displacement and the perpetrators as well as the perceived risk contexts. My concerns on the article are mainly focused on the methodology, as written below.

Response 

Thank you, we appreciate your comment.

2. Type of Qualitative Study.

Does this study fall into any specific type qualitative study, such as phenome

---

## [Decision Letter · Decision Letter 1]

20 Nov 2024

Exploring perceptions and experiences of gender-based violence among women in a refugee camp setting in Uganda - a qualitative study

PONE-D-24-21429R1

Dear Dr.  Lukasiak,

We’re pleased to inform you that your manuscript has been judged scientifically suitable for publication and will be formally accepted for publication once it meets all outstanding technical requirements.

Kind regards,

Derebe Madoro Bunte

Academic Editor

PLOS ONE

---

## [Editor Report · Acceptance letter]

22 Nov 2024

PONE-D-24-21429R1 

PLOS ONE

Dear Dr. Lukasiak, 

I'm pleased to inform you that your manuscript has been deemed suitable for publication in PLOS ONE. Congratulations! Your manuscript is now being handed over to our production team.

Kind regards, 

on behalf of

Dr. Derebe Madoro Bunte 

Academic Editor

PLOS ONE